# Extracting individual characteristics from population data reveals a negative social effect during honeybee defence

**Tatjana Petrov**[1,2]*, **Matej Hajnal**[1,3], **Julia Klein**[1,2], **David Šafránek**[3]*,
**Morgane Nouvian**[2,4,5]*

**1** Department of Computer and Information Sciences, University of Konstanz, Konstanz, Germany, **2** Centre for the Advanced Study of Collective Behaviour, University of Konstanz, Konstanz, Germany, **3** Systems Biology Laboratory, Faculty of Informatics, Masaryk University, Brno, Czech Republic, **4** Department of Biology, University of Konstanz, Konstanz, Germany, **5** Zukunftskolleg, University of Konstanz, Konstanz, Germany

* tatjana.petrov@uni-konstanz.de (TP); safranek@fi.muni.cz (DŠ); morgane.nouvian@uni-konstanz.de (MN)

## Abstract

Honeybees protect their colony against vertebrates by mass stinging and they coordinate their actions during this crucial event thanks to an alarm pheromone carried directly on the stinger, which is therefore released upon stinging. The pheromone then recruits nearby bees so that more and more bees participate in the defence. However, a quantitative understanding of how an individual bee adapts its stinging response during the course of an attack is still a challenge: Typically, only the group behaviour is effectively measurable in experiment; Further, linking the observed group behaviour with individual responses requires a probabilistic model enumerating a combinatorial number of possible group contexts during the defence; Finally, extracting the individual characteristics from group observations requires novel methods for parameter inference.

We first experimentally observed the behaviour of groups of bees confronted with a fake predator inside an arena and quantified their defensive reaction by counting the number of stingers embedded in the dummy at the end of a trial. We propose a biologically plausible model of this phenomenon, which transparently links the choice of each individual bee to sting or not, to its group context at the time of the decision. Then, we propose an efficient method for inferring the parameters of the model from the experimental data. Finally, we use this methodology to investigate the effect of group size on stinging initiation and alarm pheromone recruitment.

Our findings shed light on how the social context influences stinging behaviour, by quantifying how the alarm pheromone concentration level affects the decision of each bee to sting or not in a given group size. We show that recruitment is curbed as group size grows, thus suggesting that the presence of nestmates is integrated as a negative cue by individual bees. Moreover, the unique integration of exact and statistical methods provides a quantitative characterisation of uncertainty associated to each of the inferred parameters.

**Data Availability Statement:** The software used for the analysis and plotting is publicly available on GitHub at the repository: https://github.com/

xhajnal/DiPS. All the input and output files are publicly available at the Zenodo repository https://zenodo.org/record/6600766##.YpbF13VBw9E.

**Funding:** TP's research is supported by the Ministry of Science, Research and the Arts of the state of Baden-Württemberg. MH's research was supported by Young Scholar Fund (YSF), project no. P83943018FP430_/18. JK's research was supported by the AFF (Der Ausschuss für Forschungsfragen, EU-Anschubfinanzierung, Univ. of Konstanz). TP, MH, JK were further funded by the DFG Centre of Excellence 2117 'Centre for the Advanced Study of Collective Behaviour' (ID: 422037984). DS's research has been partially supported by the Grant Agency of Czech Republic grant no. GA22-10845S. MN's research was supported financially by the Zukunftskolleg (University of Konstanz) and by a DFG research grant (project number 414260764). The funders had no role in study design, data collection and analysis, decision to publish, or preparation of the manuscript.

**Competing interests:** The authors have declared that no competing interests exist.

## Author summary

In this paper, our interdisciplinary team has significantly improved the understanding of how honeybees coordinate their actions during defence. Our first step was to measure the output behaviour of groups of bees under controlled experimental conditions. We then developed a model and methodology that allow us to quantify how the responsiveness to the alarm pheromone evolves during a defensive event, for a given group size. We show that recruitment becomes less effective as group size increases, thus revealing the existence of a negative social effect that acts on top of alarm pheromone communication. Our contribution is thus two-fold: on the computational side, we provide new tools to extract individual characteristics from population data, which is a challenging issue in the study of collective behaviour. On the biological side, we provide evidence that bees weight in their social context when making the decision to sting. We hypothesize that this may be an important mechanism to prevent recruitment from spinning out of control, ultimately preserving the colony from workforce depletion.

## Introduction

From fish schools to ant colonies, animal collectives offer some of the most spectacular examples of decentralised organisation. Understanding how they achieve these feats is complicated by the intrinsic interconnection between individuals: each individual is influenced by the presence and actions of other group members but is also influencing them in return. The collective output that emerges from these complex interactions can thus seldom be predicted by extrapolating from the behaviour of isolated individuals. For example, weak individual preferences or memories can be strengthened by social signals [1, 2]. Interactions between group members may even result in emergent sensing abilities that only exist at the group level [3]. An individual's performance in a given task may also vary unpredictably depending on its social context, despite being consistent within each situation [4, 5]. In social insects, being in a group in itself may gate individual responses, especially when considering "altruistic" behaviours such as colony defence [6] or thermoregulation [7]. As a final example, seemingly opposite individual preferences are observed in cockroaches depending on whether they are tested in groups or alone [8, 9]. Taken together, these studies suggest that a mechanistic understanding of collective phenomena, i.e. being able to predict group behaviour from the decision rules followed by individuals, necessitates the evaluation of individual characteristics within the specific context of the group.

However, measuring the responses of individuals that are embedded in a group is easier said than done. First, identifying specific individuals among a large number of conspecifics may be challenging, especially when it is not possible to tag the animals. Second, automated tracking often requires expensive and technologically demanding arrays of recording devices and is still mostly limited to spatial positioning. On the other hand, group behaviour is usually more amenable to measurements. This is particularly true when considering tasks such as the selection of suitable shelters, resources or routes, or tasks that produce a quantifiable outcome (e.g. amount of food gathered, number of intruders repelled). The challenge, then, becomes to extract the individual characteristics that led to such collective output.

Honeybees live in densely populated nests, in which they also store resources in the form of pollen and honey. This makes their colonies very attractive troves of nutrients for many predators, including large mammals such as bears and humans. To fend them off, the bees have to band together into a collective stinging attack. This defensive reaction is typically initiated by

(transiently) specialised bees termed guard bees, who monitor the colony's surroundings. They react to large disturbances such as vertebrates by stinging the intruder or by extruding their stinger and fanning their wings, sometimes while running into the hive. In both cases, their behaviour causes the release of the sting alarm pheromone (SAP), a complex pheromonal blend carried directly on the stinger. This chemical signal arouses nearby bees and recruits them to the site of the disturbance, where they decide whether to participate or not in the defensive effort by stinging or otherwise harassing the predator [10, and references therein]. Hence the SAP plays a major role in amplifying the defensive reaction of the colony so that it reaches critical mass.

Because of the predominant effect of the SAP, the defensive behaviour of honeybees against vertebrates is typically described as a positive feedback loop in which the more bees are stinging, the more they release the SAP and hence the more new bees are recruited into stinging. Thanks to this mechanism the bees can quickly mount an effective defence against intruders, which is of vital importance for the colony. However, defending also has a cost: the defenders may get injured or die while fighting, resulting in the loss of colony workforce. This is especially true when considering the stinging behaviour of honeybees against vertebrates because their barbed stinger remains embedded in elastic skin and tears off from the bee's abdomen, causing the bee to die from the injury. In order to balance between achieving an efficient defence and preserving workforce, we expect that the decision to sting is tightly regulated at the individual level. In particular, we postulate that bees consider more social information that just SAP levels, and that these social cues provide a negative feedback that counteracts SAP recruitment. With this study, our biological aim is thus to quantify the effect of SAP levels on the likelihood to sting of individual bees, in given social contexts. We focus on group size as previous studies have found that this factor can influence aggressive responses in social insects [11–14].

To do so, we first observed the behaviour of groups of bees confronted with a fake predator (a rotating dummy) inside an arena and quantified their defensive reaction by simply counting the number of stingers embedded in the dummy at the end of a trial. Second, we propose a mathematical model of the group dynamics, which transparently links the probabilistic choice of a single bee to sting at a given alarm pheromone concentration, to the collective outcome observed in the experiment. Concretely, each honeybee is modelled as a Markovian agent potentially triggered into stinging at a given alarm pheromone concentration, and which releases more alarm pheromone upon doing so. Each stinging bee thus modifies the environment so that more bees may be triggered into stinging, leading to a chain of reactions that stops when no additional bee is recruited (steady-state reached). The model of a group is formalised as a multi-dimensional discrete-time stochastic process, fully parametrised by a series of parameters $r_0, r_1, \ldots, r_{n-1}$, where $n \geq 1$ is the group size. The parameter $r_k$ represents the probability that a single bee decides to sting when $k$ other stings have occurred. Our goal becomes to infer these parameters from the experimental observations.

While there exists a rich body of work on parameter inference for population Markov models, these techniques are typically considering experimental observations over time (time-series data), and not only at steady-state. To this end, we propose a unique methodology that combines formal methods for parameter synthesis and probabilistic model checking for parametric Markov chains [15–17], which allows us to compute the expected frequency of each of the possible experimental outcomes (i.e. the likelihood function) in the form of a polynomial expression over parameters $r_0, r_1, \ldots$ [18]. Once the analytic form of likelihood functions are obtained, we subsequently apply the standard statistical procedures for parameter search [19, 20] to find the parameters agreeing with data, and, in addition, to quantify the uncertainty of the inferred values. The model, together with this methodology, allows us to successfully predict how the SAP being released by attacking bees influences their nestmates, and therefore to

extract the behaviour of individuals from the population data. The methodology is based on two essential steps. First, the hypothesised likelihood functional forms are fitted to the experimental data. An important aspect of our methodology is that the techniques employed are agnostic: they do not require any assumptions about the functional form that parameters $r_0$, $r_1$, . . . follow. Second, model selection is performed between two biologically plausible hypotheses. In particular, we explore whether the parameters follow either a linear or a sigmoidal trend [21].

Thus, an important contribution of this paper is methodological and can be summarised in the following points: (i) we establish a new mechanistic model hypothesising the behaviour of an individual based on expert knowledge (decision to sting and the likelihood in given physical and social context); (ii) the model allows to predict stochastic dynamics of the entire population emerging from the behaviour of individuals, for a range of group sizes; (iii) the proposed methodology allows to automatically infer model parameters (reflecting the behaviour of an individual) from data collected at the steady-state of the entire population; (iv) the unique integration of exact and statistical methods provides a quantitative characterisation of uncertainty associated to each of the inferred parameters. The described methodology is available in the form of a versatile software—DiPS (Data-informed Parameter Synthesis for Discrete-Time Markov Chains, https://github.com/xhajnal/DiPS) combining multiple methods for parameter inference for Discrete-Time Markov Chains utilising state-of-the-art tools, PRISM [22], Storm [23], z3 [24], dreal [25], and scipy [26].

With this methodology, we obtained results consistent with previous findings on how stinging likelihood varies depending on the SAP concentration, based on individual measurements [21]. Furthermore, we validated our prediction by demonstrating that individual bees become less likely to sting as the group grows larger, for any given SAP concentration. Our interpretation is that larger groups provide more negative cues, hence curbing recruitment from the SAP. The resulting "diffusion of responsibility" may be an important mechanism for the colony to successfully balance between defence and other tasks.

## Materials and methods

### Experimental data

The experiment was replicated three times. The first replicate was performed at the University of Otago (New-Zealand), on a single colony. The sample size was 60 groups of bees for each of the 4 group sizes tested (1–2-5-10). The second and third replicates were done at the University of Konstanz (Germany) on a total of 6 colonies. The second replicate (2019) included 4 colonies, each contributing equally to the dataset. We tested 6 group sizes (1–2-5-7–10-15), and the final sample sizes were 68, 68, 60, 56, 52 and 48 groups of bees respectively. The last replicate (2020) spanned again 4 colonies, including 2 from the previous summer. Each colony contributed 10 groups of bees to the 4 group sizes tested (1–2-5-10), hence a final sample size of 40 groups per data point. This information is summarized in Table 1. To present the methods, we only focused on a single group size (10 bees) with data pooled from the 2nd and 3rd replicates (in bold in Table 1). Thus our final sample size, $N$, is 92 groups for these sections. We show the result distribution for this data in Section Experimental data with groups of 10 bees, and for all datasets in S1 Text (Section Experimental Data).

Defensive bees were collected by waving a black ostrich feather in front of the colony, as described previously [14]. They were then sealed into a plastic bag, chilled and placed in groups of 10 into modified syringes with *ad libitum* sugar water (50% vol/vol). Note that within one group, all bees came from the same colony. After recovering for at least 15 min, they were then tested for their aggressive behaviour. The protocol for the aggression assay itself

**Table 1. Summary of the experimental data collected.**

| Data set | Location | Colonies | Test duration | Group sizes | Sample sizes |
|---|---|---|---|---|---|
| 1 | Dunedin (NZ) | A | 3 min | 1–2–5–10 | 60–60–60–60 |
| 2 | Konstanz (Ger) | B-C-D-E | 10 min | 1–2–5–7-**10**-15 | 68–68–60–56-**52**-48 |
| 3 | Konstanz (Ger) | B-C-F-G | 10 min | 1–2–5-**10** | 40–40-**40**-40 |

has also been described in detail in [14]. Briefly, the bees were introduced into a testing arena where they faced a rotating dummy coated in leather. Stinging behaviour was scored by counting the number of stingers embedded in the dummy at the end of the test. The test duration for replicate 1 was 3 min, as in previous work [14]. In replicates 2 and 3, we increased this duration to 10 min to make sure that all bees had enough time to sting following the build up of alarm pheromone. Pilot experiments indeed showed that very few bees would sting after this time even in large groups. Nonetheless, the data from the 1st replicate is still included in this paper because the results are qualitatively the same. As an outcome of these experiments, we thus have a measure of the frequency at which each specific number of stinging bees was observed, for each group size.

## Probabilistic model of collective stinging behaviour

In order to unravel how each individual honeybee adapts its stinging behaviour during the course of a defensive event, we created a biologically-relevant mathematical model linking individual responses to the group dynamics. The model accounts for variability in aggressiveness among bees and is parametrised by the set of parameters representing probabilities to sting of an individual bee at each alarm pheromone concentration level. Based on these parameters, our model thus predicts the pattern of stinging responses observed for a given group size.

Mathematically, the model is represented as a discrete-time Markov chain (DTMC) with parametrised probabilities on the edges (a parametric Markov chain, pMC) (In our case study, we deal with a special case of a DTMC without cycles, also called a branching process.). Formal definitions of objects used throughout this paper, such as Markov chain, parametric Markov chain, bottom strongly connected component (BSCC), can be found in S1 Text (Section Preliminaries). In the following paragraphs, we gradually describe the model starting from the considered biological assumptions and following with a technical explanation of the model. The model is first described from the perspective of an individual bee. In the next step, it is shown how the population of two bees is handled. Finally, we show how the model can be (in an automated way) extended to a population of arbitrary size.

**Modelling assumptions.** The model reflects several simplifying assumptions that comply with the current biological knowledge and our experimental conditions. The model provides the link between individual stinging probabilities and the probability of attaining a specific experimental outcome for a collective. To that end, adequate assumptions on the collective behaviour of bees in time and space have to be considered.

The environment is considered without the process of pheromone degradation because the pheromone is released within an arena with a fairly stable atmosphere. The temporal behaviour of the population is implemented in discrete steps where every change of state corresponds to a respective change of the environment affected by the released alarm pheromone (in response to stinging events).

The model entirely abstracts from the spatial characteristics of the population. In particular, spatial homogeneity in our context is justified because the alarm pheromone is known for its

                                

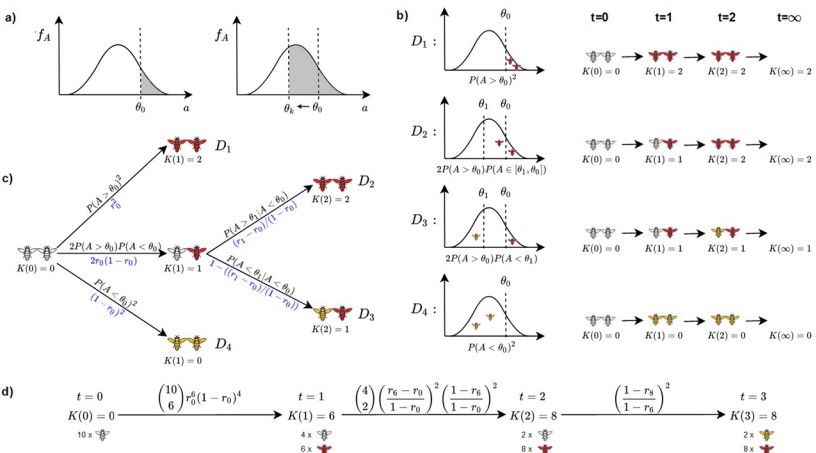

**Fig 1. Modelling stinging response for *n* = 2.** A) Probability density function of aggressiveness. B) Four different situations for a group of two bees. Simulation of different stinging behaviour across time for four different initial aggressiveness situations. C) A Markov chain model of four different stinging scenarios for a group of two bees. D) An example trace of a Markov chain model generated for *n* = 10.

high volatility, the arena is quite small, and the rotating dummy constantly mixes the air. Hence, since we consider bees equally influencing each-other, any permutation of the population is identical. Furthermore, we often observe very fast responses from bees at the periphery of the arena, especially after the 1st sting (within seconds), supporting the idea that the alarm pheromone is dispersed quickly and that multiple bees may react simultaneously to it.

Given the fast reaction times observed experimentally, we assume that any bee stings immediately when the stinging condition is met. Note also that bees lose their stinger upon stinging a vertebrate predator or the leather-coated dummy in our arena, which is why we consider that bees can only sting once. We also assume that multiple bees may decide to sting at the same time step (rather than always one-by-one)—this goes with the fact that the environment is changing globally for all the bees in the arena.

**Stinging behaviour of an individual bee.**   Denote by $\mathbb{N}$ the set of naturals and by $[a..b] \subseteq \mathbb{N}$ the interval set $\{a, a + 1, \ldots, b\}$. The set of real numbers will be denoted by $\mathbb{R}$. We model the aggressiveness of a single bee indexed by $j \in [1..n]$ in a group of size $n \in \mathbb{N}$ by a random variable $A_j \in \mathbb{R}_+$, distributed according to a probability density function (pdf) $f_A$ (in Fig 1A, we represented $f_A$ by a shape of a normal distribution, yet our modelling approach does not actually need to assume a shape for $f_A$.). Then, a single bee makes a decision to sting if its aggressiveness exceeds the predator-specific threshold, i.e. $A_j > \theta_0$, where $\theta_0$ represents how threatening the predator is evaluated from its characteristics. Once the stinging process is triggered, any remaining bee decides to sting if $A_j + \Delta_k > \theta_0$, where $\Delta_k$ is the respective aggressiveness increase (assumed to be equal for each bee in the colony), after $k$ bees have stung and hence the concentration level of the pheromone inside the arena is $k$ (since each stinging bee releases exactly one unit of alarm pheromone). Introducing notation $\theta_k := \theta_0 - \Delta_k < \theta_0$, allows us to rewrite the stinging condition as $A_j > \theta_k$. Notice that rewriting the stinging condition in this form (stinging condition $A_j > \theta_k$) may suggest an interpretation that the aggressiveness is fixed and the stinging threshold changes in response to the concentration level $k$. Such an interpretation is different from the one the model is originally built upon (stinging condition $A_j + \Delta_k > \theta_0$), that aggressiveness changes with stinging events, as this is the known action of the alarm pheromone at a large range of concentration levels [21]. Since the two

                                        

interpretations are mathematically equivalent, for the purpose of presentation clarity, we will further represent the stinging condition by $A_j > \theta_k$.

The probability that any bee stings prior to alarm pheromone release equals $P(A_j \geq \theta_0) = \int_{\theta_0}^{\infty} f_A(x)dx$, illustrated as the shaded area right from the threshold $\theta_0$ in Fig 1A. We denote by $r_k$ the stinging probability at pheromone concentration level $k$:

$$r_k := P(\text{Bee } j \text{ stings at context } k) = P(A_j > \theta_k) = \int_{\theta_k}^{\infty} f_A(x)dx,$$

visualised as the shaded area right from the threshold $\theta_k$ in Fig 1A. Since the aggressiveness increases with the alarm pheromone, it will be assumed that $\theta_0 \geq \theta_1 \geq \ldots$, and, consequently, the probability $r_k$ is a non-decreasing function in $k$. Our goal is to infer the values of $r_0, r_1, \ldots$, the stinging probability of a single bee for varying alarm pheromone concentration levels. Notice that direct measures of $r_k$ for different values of $k$, by repeated controlled experiments, are possible [21], but only with isolated bees. The more naturalistic social context of a group may change the stinging responses of bees (we presented many examples of such effect in the introduction). This is why, in this study, we took a different approach: rather than externally controlling the alarm pheromone concentration, we take advantage of it being released by stinging bees and evaluate its effect by observing the variation in end-point behaviour between groups of bees exposed to the same initial stimulus.

**Stinging behaviour of two bees.** Let $K(t)$ be the random variable modelling the alarm pheromone concentration at step $t \in \mathbb{N}$. We denote by $K(\infty)$ the value measured in the experiment which is the number of stinging bees at the end of the test. In Fig 1B, we illustrate the three possible experimental outcomes for $K(\infty) \in \{0, 1, 2\}$, arising when two bees are picked from a colony:

(i). $A_1 > \theta_0$ and $A_2 > \theta_0$: both bees respond to the visual stimulus and decide to sting independently; This situation occurs with probability $r_0^2$.

(ii). $A_1 > \theta_0$ and $A_2 \in [\theta_1, \theta_0]$: bee 1 responds to the visual stimulus and stings, bee 2 gets recruited to sting because of the unit of alarm pheromone released by bee 1; This situation occurs with probability $2r_0(r_1 - r_0)$ (where the factor 2 accounts for a symmetric situation where $A_1 \in [\theta_1, \theta_0]$, $A_2 > \theta_0$, considered equivalent in our model);

(iii). $A_1 > \theta_0$ and $A_2 < \theta_1$: bee 1 responds to the visual stimulus and stings, bee 2 remains non-aggressive despite the alarm pheromone released; This situation occurs with probability $2r_0(1 - r_1)$.

(iv). $A_1 < \theta_0$ and $A_2 < \theta_0$: neither bee is responsive to the visual stimulus; This situation occurs with probability $(1 - r_0)^2$.

It is important to note that the same final observation can be attained via different paths: situations (i) and (ii) both lead to a total of 2 bees stinging, but understanding the role of the alarm pheromone requires considering both ways. We can express the probabilities of each of the four situations in terms of parameters $r_0, r_1, \ldots$. In summary, the experimental outcome will follow the following distribution:

$$K(\infty) \sim \begin{pmatrix} 0 & 1 & 2 \\ r_0^2 + 2r_0(r_1 - r_0) & 2r_0(1 - r_1) & (1 - r_0)^2, \end{pmatrix}, \tag{1}$$

under the interpretation $r_0 = P(A > \theta_0)$ and $r_1 = P(A > \theta_1)$.

For the case of two bees, analysing all possible paths is possible to do by hand. However, for $n \geq 3$ generating all possible paths and computing the respective probabilities becomes

                    

challenging. For this reason, we introduce a DTMC abstraction that mimics the stinging process: for example, the four stinging paths listed above will be the traces of the DTMC, shown in Fig 1C. Such DTMC helps us to efficiently and automatically enumerate and compute the probabilities of each of the four paths. The observable state at the time point is $K(t)$, while the internal state tracks the aggressiveness in relation to changing thresholds. Such internal representation allows us to correctly assign the respective transition probabilities. For instance, in Fig 1C, the state depicted with one red and one white bee has the internal state $\{ > \theta_0, <\theta_0\}$ and an observable $K(1) = 1$, because one bee has stung within the first step. Then, the successive stinging event will depend on whether the second bee's aggressiveness level surpasses $\theta_1$. If this is the case, the trace moves to state $q' = \{> \theta_0, >\theta_1\}$, occurring with probability $P(A > \theta_1 \mid A < \theta_0) = \frac{r_1 - r_0}{1 - r_0}$, which is well-defined thanks to the assumption that values of $r_0$, $r_1$, ... are non-decreasing.

Finally, after the DTMC is constructed, it becomes possible to compute the function $K(\infty)$ as a function of parameters $r_0$, $r_1$, ..., and finally to infer these parameters based on the empirical estimates of $K(\infty)$. The inference methods are presented in Section Parameter inference.

**Stinging behaviour in a population.** A generalisation for the case of $n \geq 3$ bees is straightforward. The relation of initial aggressiveness $A_1$, ..., $A_n \sim f_A$, with respect to thresholds $\theta_0, \theta_1$, ..., will determine the stinging response over time, and induce variability on the final observation $K(\infty)$.

Detailed construction of the DTMC for population of any given size is given in S1 Text (Section Model). In Fig 1D, we show one possible trace for a group of 10 bees: each state is represented as a multi-set of labels indicating our knowledge of aggressiveness in relation to thresholds. For instance, (the internal) label $\{6(> \theta_0), 2(> \theta_6), 2(< \theta_6)\}$ denotes a state with 6 bees who sting in response to the visual stimulus (before any alarm pheromone is released), two bees that sting after the first step, and two bees not stinging after the first step. The observable label of this state is $K(1) = 6$.

## Parameter inference

Parameter inference for Markov (population) models is a long-studied problem, relevant in a wide range of applications including systems biology [27], finance [20], and beyond. However, the existing state-of-the-art techniques are typically designed for the case when time-series experimental observations are available. We instead deal with measurements only at steady-state, yet these steady-state measurements can be repeated under controlled experimental conditions. We propose a novel methodology which successfully exploits this possibility of assessing the frequency at which each steady-state is reached. Parameters in the parameter inference problem we deal with are identifiable (possible to infer from the available data unambiguously) due to a model feature: since a number of different experimental outcomes can be observed at the steady-state (the model contains more than one observable outcome), a large enough number of repeated measurements allows us to constrain the parameter space with the data sufficiently.

The key step in parameter inference is the approximation of the likelihood function. The likelihood function for steady-state observations in a parametric Markov chain (pMC) is not available in analytical form. Employing approximate forms of likelihood increase the uncertainty involved in the inference algorithm (in addition to the limited sample size and the hyperparameters of a Bayesian inference scheme, such as prior distributions, number of perturbation kernels, simulation length). We instead employ formal methods to obtain the exact likelihood for given data in terms of rational functions over parameters of the pMC, in order to reduce uncertainty and improve the scalability of the parameter inference. We first recast

the data observations into a set of temporal properties, Probabilistic Computational Tree Logic [28] in our case, and leverage the parametric model checking tools, PRISM [29] or Storm [23], to obtain the rational functions that exactly characterise the reachability of respective terminal states. Subsequently, we define and implement methods employing these rational functions to:

(i) efficiently infer parameter points closest to data observations via maximum likelihood, (ii) quantify uncertainty in a Markov chain Monte Carlo (MCMC) parameter inference scheme.

**Rational functions as symbolic expressions for measured properties.** In the example with two bees, the distribution among the three possible terminal states can be captured by the polynomials shown in Eq 1. In the respective DTMC, shown in Fig 1C, there are three different terminal states (the first one being equivalently the state reached due to distributions $D_1$ and $D_2$, and with observable $K(1) = K(\infty) = 2$, the second one being $D_3$, and the third one $D_4$). These are technically termed *bottom strongly connected components* (BSCCs, see S1 Text Section Preliminaries, Definition BSCC). For a general DTMC with multiple BSCCs, we derive the distribution among the BSCCs as polynomial expressions over model parameters $r_0, r_1, \ldots$, through the following encoding:

$$\text{let } f_k(\mathcal{V}) \in \mathsf{Pol}(\mathcal{V}) \text{ be such that}$$
$$\text{for all } \theta \in [0,1]^{|\mathcal{V}|}, \quad f_k(\theta) = \mathsf{P}(\mathcal{M}(\theta) \models FG(B_k)),$$

that is, $f_k(\mathcal{V})$ is a rational function over variables $\mathcal{V} = \{r_0, r_1, \ldots\}$, exactly characterising the reachability of a BSCC uniquely labelled with $B_k$ in a parametric Markov chain $\mathcal{M}_{\mathcal{V}}$. We omit subscript $_{\mathcal{V}}$ when clear from the context. For instance, in the example of two bees, we have $f_2(r_0, r_1) = (1 - r_0)^2$, where $B_2$ is the BSCC with both bees non-stinging (denoted by $D_4$ in Fig 1). Note that, in general, the temporal formula $FG(B_k)$ specifies the behaviour of eventually reaching a BSCC $B_k$. Probabilistic model checking technology allows to evaluate the probability of satisfaction of this formula as a function of parameters of the Markov chain. In the implementation, we leverage existing model checking tools PRISM [29] and Storm [23] to obtain these polynomials.

**(i) Optimisation.** The values of parameters are found, such that the rational functions are closest to the data observation (in terms of least squares distance (L2)). To evaluate the uncertainty related to each of the inferred parameter values, we apply sampling-based local sensitivity analysis—see S1 Text (Section Sampling-Based Local Sensitivity Analysis).

**(ii) Sampling-based inference with exact likelihood.** We implement a basic Metropolis-Hastings scheme [30], a Markov chain Monte Carlo algorithm, where we employ the knowledge of likelihood (rational) functions pre-computed with model checking to evaluate the likelihood in each newly sampled parameter point. Starting in a selected initial point $\theta_{init}$, Metropolis-Hastings walks in the parameter space for a selected number of iterations. In each iteration, a *transition function* picks a new point $\theta'$ in the parameter space by perturbing the current point $\theta$ with an adjustable variation value. Next, likelihoods of these two points, $\theta$ and $\theta'$, are compared, and if the likelihood of the new point is larger $P(D_{obs}|\theta') > P(D_{obs}|\theta)$, we *accept* the proposed point and move in the parameter space. If the likelihood is smaller, there is a small probability of accepting the new point, $\theta'$—this helps to avoid local optima. Lastly, if the proposed point is *rejected*, we select the current point, $\theta$, for the next iteration. The set of accepted points is used to approximate the posterior distribution. For more dimensions, a scatter-line plot showing each of the accepted points is created—see the results in Section Parameter inference.

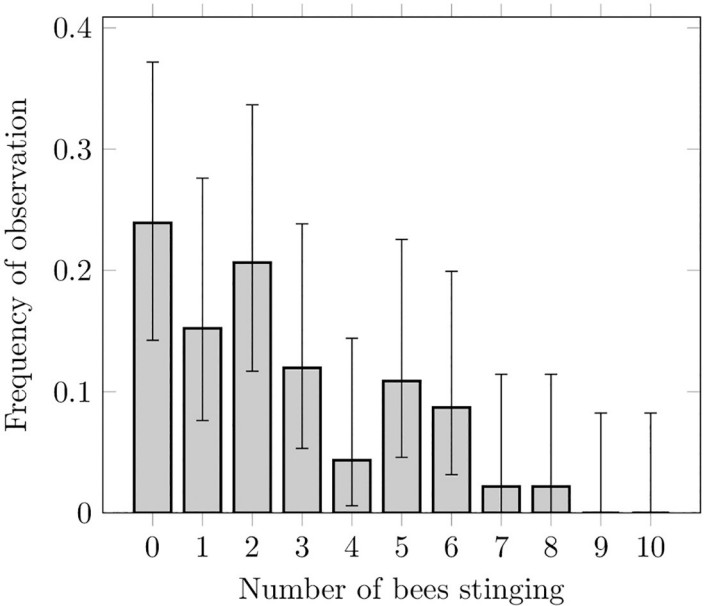

**Fig 2. Experimental data with Agresti-Coull confidence intervals (using Dunn's correction) and 90% confidence level.** Frequencies of the number of stinging bees (22, 14, 19, 11, 4, 10, 8, 2, 2, 0, 0) resulting from 92 repeated experiments.

## Model selection

We are ultimately interested in understanding how the stinging probabilities change with the pheromone concentration level. Since there are two shapes observed in experiments with single bees [21], we perform model selection between two biologically plausible hypotheses—that the parameters follow either a linear or a sigmoidal trend. While the first one amounts to that each pheromone concentration unit equally changes the stinging probability, the latter suggests that, for each individual, there exists a phase transition between a low and high stinging probability.

While understanding whether the trend is linear or sigmoidal is interesting as it enjoys a natural biological interpretation, it also simplifies the inference process. To this end, it is worth mentioning that the model with ten honeybees involves ten different parameters $r_0, r_1, \ldots, r_9$. The statistical significance for estimating functions $f_0, f_1, \ldots f_{10}$ from our given experimental data—92 repeated samples falling into one of eleven categories—is represented in form of confidence margins in Fig 2. While these margins may be satisfactory for inferring $f_0, f_1, \ldots, f_{10}$, the propagation of uncertainty to each of the model's parameters $r_0, r_1, \ldots, r_9$ renders inference unsatisfactory for some of the parameters. Concretely, the uncertainty with respect to the parameters can be read-out from the Metropolis-Hastings results, and it confirms the intuition that the parameter $r_0$ can be estimated with lowest uncertainty, and parameter $r_9$ with highest. To intuitively explain, consider first the case of two bees. Our data tells us more about parameter $r_0$ than about $r_1$: the information obtained from the experiment where both bees immediately sting does not tell us anything about the parameter $r_1$ (the probability to sting when one amount of pheromone is present). At the same time, each of the outcomes tells us something about parameter $r_0$. Indeed, the parameter $r_0$ plays a role in each of the expressions shown in Eq 1 (while the parameter $r_1$ does not play a role in outcome $K(\infty) = 2$). In case of ten bees, strikingly, the parameter $r_9$ will be estimated from only those stinging cascades, where at least

nine bees have stung, an event with very low likelihood (concretely, $(1 - r_0)^9$). Hence, for the above reasons, it is desirable to infer the trend of the stinging response, while accounting for the associated uncertainty, instead of inferring each of the parameters separately.

**Linear model.**   The linear model is depicted as a simple linear dependence of parameter values. This linear shape of $r_i$ values can be expressed as $r_i = r_0 + i \cdot \Delta$ This transformation decreases the number of parameters to two: $r_0$ and $\Delta$.

**Sigmoidal model.**   The sigmoidal model is depicted as a dependence of parameter values using Hill function. Hence, the value of respective $r_i$ can be expressed as: $r_i = r_0 + \frac{V_{max} - r_0}{1 + \left(\frac{K_m}{i}\right)^n}$. This transformation decreases the number of parameters to four: $r_0$—basal level, $V_{max}$—saturation level, $K_m$—value at which the hill function is at half of the slope, and $n$—Hill coefficient indicating the slope of the curve.

**Model selection.**   To compare the models and select the best one we compute the Akaike Information Criterion (AIC) [31] that is a measure of the relative goodness of fit. It weighs the fit of the model against its complexity, measured by the number of independent parameters. We use the following formula for the AIC based on the least squares fitting of the model:

$$AIC = n \, \log\left(\frac{\mathrm{RSS}}{n}\right) + 2k, \tag{2}$$

where $n$ is the number of observations, RSS the residual sum of squares, and $k$ the number of free parameters. The model with the lowest AIC is considered the best one [32].

## Tool implementation and reproducibility of results

For the parameter inference part we have used our DiPS tool version 1.27.4 and version 1.21 for adapted version of Optimisation and Metropolis-Hastings for non-decreasing parameters. The model selection part is implemented in a short R script. When sourced, it automatically runs all analyses, outputs the results in the console, and saves the according plots on the machine.

The whole analysis has been run on Skadi: Ubuntu 18.04, i9–9900K, 32GB RAM, SSD. README file describing the analysis in a step-by-step manner, input files, model, properties, data, scripts, and the full set of results are available at Zenodo repository at https://zenodo.org/record/6600766#.YpbF13VBw9E.

## Results

### Experimental data with groups of 10 bees

Groups of 10 bees were placed inside a small arena, in which they confronted a rotating dummy. After 10 min, the number of stingers embedded in the dummy were counted. This procedure was repeated 92 times, such that the frequency at which a given number of stinging bees is observed could be estimated. This data, presented in Fig 2, thus provides a measure of the steady-states reached by our system.

### Parameter inference

In this section we present the result of the parameter inference for the model of 10 bees using two different methods: optimisation (equivalent to maximum likelihood in our case), and Metropolis-Hastings (Bayesian inference via MCMC). Both of the techniques utilise the analytic forms of data likelihood functions, precomputed via formal methods. Besides the *agnostic model* with a non-decreasing constraint on the parameter values, we propose two shapes of the parameter values—*linear* and *sigmoidal*—defined in Section Model selection.

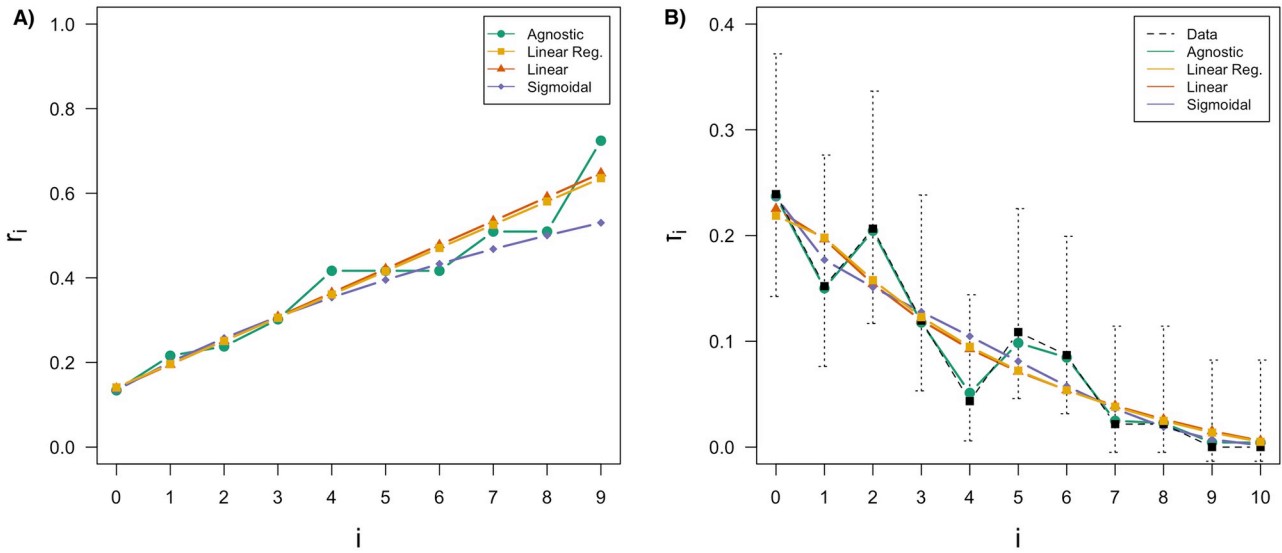

**Fig 3. Single point estimation (parametrisation minimising the L2 distance between the rational functions and data).** (A). For comparison, rational function values (coloured lines), data (dashed line), and 90% confidence intervals computed from data points (black error bars) (B). On each graph, the results are shown for the agnostic model (green), a linear regression on the agnostic points (yellow), the linear model (red) and the sigmoidal model (blue).

In Fig 3, we show the parameter values inferred via optimising the L2 distance, obtained as a single parametrisation. We also visualise the posterior distribution as the result of Metropolis-Hastings accepting only non-decreasing values in Fig 4, while the setting of the method is shown in Table 2. Notice that the distribution of later parameter points (right side) is broader, reflecting higher uncertainty on these points. Indeed, these parameters are present in fewer rational functions and linked to infrequent events.

We perform model selection by comparing the AIC score for the linear model with the AIC score for the sigmoidal model. Three model variants are considered: model fit based on (i) rational function values $f_i$, (ii) parameter values $r_i$, (iii) weighted parameter values (where weights account for the relative sensitivity of parameters). The linear model has a lower AIC score for all three model variants and is therefore considered the best model explaining the data using the fewest possible parameters. Detailed computations can be found in S1 Text (Section Model Selection). Consequently, the hypothesis that aggressiveness increases linearly with pheromone level is supported by our experimental data. This finding can simplify the inference for bigger group sizes and allow easier comparisons of how this may vary with respect to group size or composition.

Note that the AIC score measures only the relative quality of the model, and is used to decide between models. Hence, we need to validate the absolute quality of the chosen model, the linear model. After inspecting the residuals of the linear model (see Fig F in S1 Text) to confirm their normality, we test the model's predictions. The coefficient of determination, $R^2$, is computed as a summary measure of the predictive power of the linear model. To check if the linear model fits the data well, we compute

$$R^2 = 1 - \frac{RSS}{TSS}, \tag{3}$$

where TSS is the total sum of squares, proportional to the variance of the data.

Our results indicate that 92.5% of the variance of the dependent variable in the data can be explained by the variance of the independent variable. Considering normalized residuals (weighted distances), even 97.3% of the variance can be explained by the linear model. S1 Text (Section Model Selection) contains detailed computations.

In conclusion, the AIC criterion showed that the linear model is better relative to the sigmoidal model. Model validation confirmed that the data fits well the linear model, since the residuals seem random and the $R^2$ scores are high.

## Application: The effect of group size on stinging behaviour

In the previous sections, we focused on a single group size to describe and establish our methods. With this done, we can now explore how group size affects the defensive behaviour of individual bees. We collected 3 datasets in total, described in the Methods and in Table 1. Note that the main differences are that the test duration was increased from 3 to 10 min after the first set of experiments to ensure that all bees had time to sting, and that more group sizes were tested in the second set. We include dataset 1 nonetheless because the results are consistent with the other replicates, suggesting that 3 min was already enough to capture most of the stinging events.

In these experiments, we varied group size from 1 to 15 bees, and again counted the number of stingers left in the dummy at the end of the test duration. The raw data distribution can be seen in S1 Text (Section Effect of Group Size). Within this range of group sizes, we could use all 3 models (agnostic, linear and sigmoidal) in order to compute the dose-response curve to the alarm pheromone of the bees embedded in each social context (Fig F in S1 Text). Based on AIC scores, we found that the linear model was again the best fit for our data (Table B in S1 Text) thus this is what we used for further analysis, but note that we obtained similar results with a linear regression on the parameter values estimated by the agnostic model. In all 3 datasets, we found that the probability to sting at a given alarm pheromone concentration decreased with increasing group size (Fig 5). When examining how the slopes and intercepts varied (Fig 6), we found indeed that the slope of the dose-response curve was significantly anti-correlated with group size in dataset 2 (Pearson's r test; $\rho = -0.8315731$, $p = 0.0404234$). A similar trend was observed in datasets 1 and 3, although it was not significant likely due to the

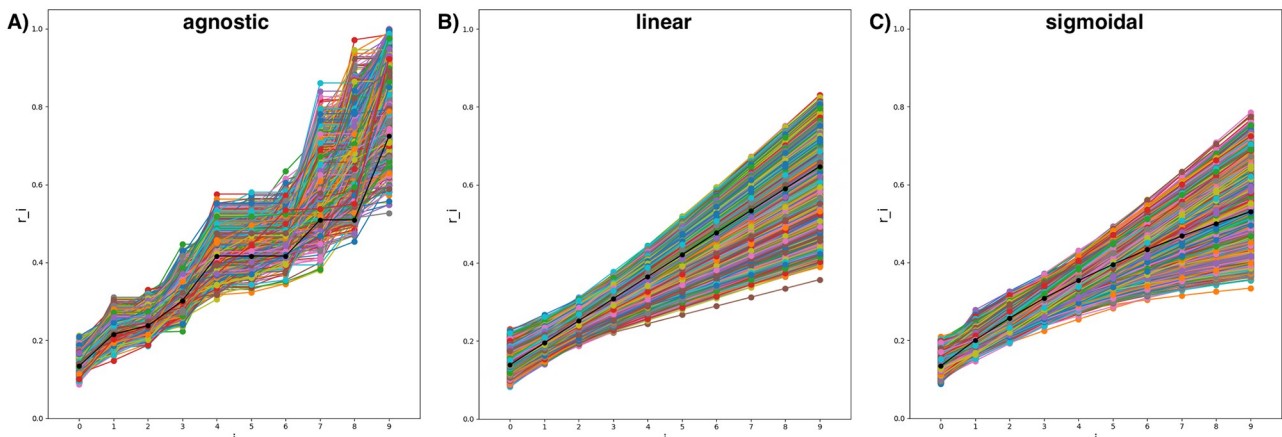

**Fig 4. Metropolis-Hastings results of the agnostic (A), linear (B), and sigmoidal (C) model: Set of accepted points.** Each accepted point shown as a line with values of respective parameter point. Burn-in period selected as 25%. We run the agnostic model for twice many iterations to check the convergence of the method. The black line shows the respective optimised point.

**Table 2. Individual settings and corresponding results achieved with the Metropolis-Hastings method for agnostic, linear, and sigmoidal model (shown in particular rows).** The columns display the following information respectively: the total number of points explored; the number of accepted points; the proportion of accepted points which were trimmed out from the beginning; the initial parametrisation; the computation time (in hours); and the number of dimensions of the explored parameter space (the number of explored parameters).

|  | # iterations | # accepted points | burn-in period | initial parameter | computational time | # parameters |
|---|---|---|---|---|---|---|
| Agnostic | 10,000,000 | 2,249 | 25% | [0.1, . . ., 0.1] | 15.67h | 10 |
| Linear | 30,000,000 | 546,934 | 25% | [0.5, 0.5] | 12.56h | 2 |
| Sigmoidal | 358,287 | 19,841 | 25% | [50, 5, 5, 0.5] | 96h | 4 |

lower number of group sizes tested (1: $\rho = -0.9332609$, $p = 0.1169501$; 3: $\rho = -0.9426515$, $p = 0.1083237$). These results thus demonstrate that alarm pheromone recruitment is curbed by the presence of nestmates. We also observed a similar pattern when looking at the intercepts: they decrease significantly with group size in dataset 2 ($\rho = -0.8248105$, $p = 0.04283618$) but not in datasets 1 and 3 (1: $\rho = 0.7495004$, $p = 0.7697062$; 3: $\rho = -0.730198$, $p = 0.2394278$). In addition, it is worth noting than our agnostic model provides a very robust estimate for the 1st parameter value r0 (since it only relies on the frequency of no attacks, which is frequently sampled). When we tested these values directly rather than the intercepts, we again found that they decreased with group size in dataset 2 ($\rho = -0.8208478$, $p = 0.02263414$), but not 1 and 3 (1: $\rho = 0.2101355$, $p = 0.6050677$; 3: $\rho = -0.4074521$, $p = 0.296274$). Thus, we conclude that attack initiation (which is based solely on visual and tactile cues rather than the alarm pheromone) may also be negatively affected by social cues from nestmates.

## Discussion

In this study, we present a probabilistic model describing the stinging response of a group of bees, allowing in-depth analysis of this collective phenomenon, based on a simple experimental procedure. Our model is straightforward in its logic, and reflects the current knowledge on the defensive strategy of honeybees. First, it considers that individual bees can vary in their responsiveness to a given threatening stimulus: this is known to arise from genetic differences among different patrilines within a colony [33] as well as from the age-dependent division of labour between worker bees [34, 35]. Second and foremost, our model includes recruitment via the sting alarm pheromone (SAP), released upon stinging, a fact that was established

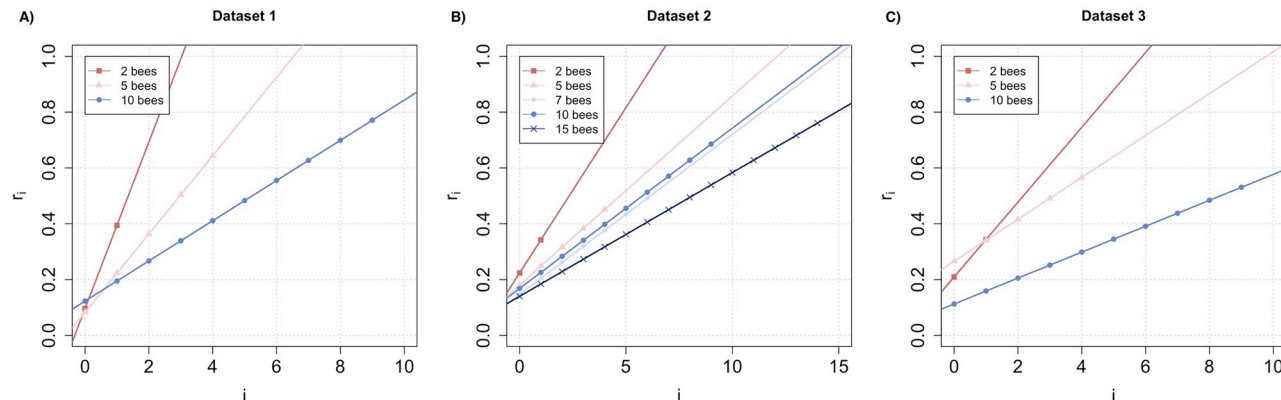

**Fig 5. Likelihood to sting as a function of alarm pheromone units, based on optimisation of the parameter points with a linear model.** The optimisation was run separately for each group size, for the 3 datasets available.

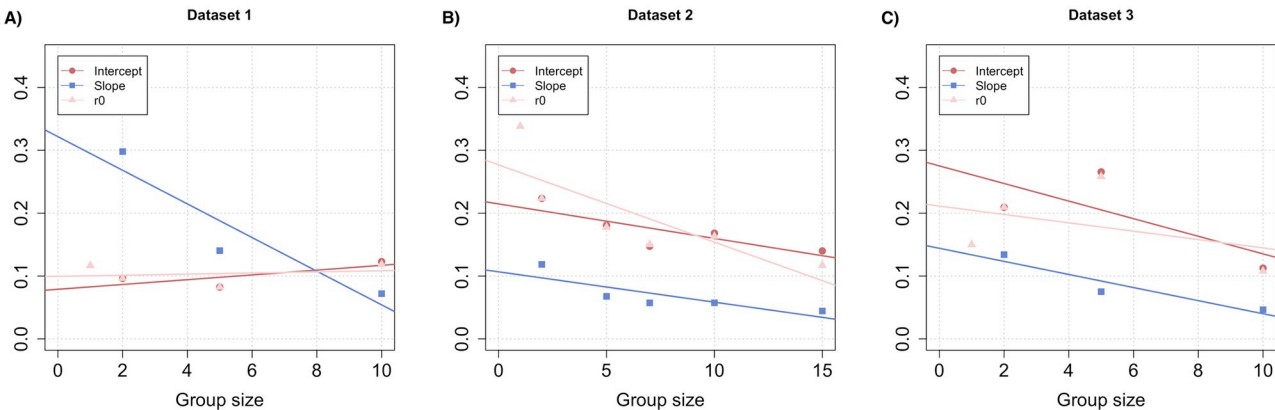

**Fig 6. Slopes, intercepts and r0 values of the alarm pheromone dose-response curve as a function of group size, for all 3 datasets.** The slopes and intercepts are based on the linear model, while the r0 value is estimated from the agnostic model. Pearson's r test; dataset 1 (A): slopes $\rho = -0.9332609$, $p = 0.1169501$, intercepts $\rho = 0.7495004$, $p = 0.7697062$, r0 $\rho = 0.2101355$, $p = 0.6050677$; dataset 2 (B): slopes $\rho = -0.8315731$, $p = 0.0404234$, intercepts $\rho = -0.8248105$, $p = 0.04283618$, r0: $\rho = -0.8208478$, $p = 0.02263414$; dataset 3 (C): slopes $\rho = -0.9426515$, $p = 0.1083237$, intercepts $\rho = -0.730198$, $p = 0.2394278$, r0 $\rho = -0.4074521$, $p = 0.296274$.

decades ago [10, 36]. More recently, it was also demonstrated that for lone bees, the efficacy of this recruitment depends on the SAP level [21]. Thanks to the computational methods that we developed in order to fit our model to experimental data, we confirmed this finding for groups of bees. This is not trivial, as the social context can have a strong influence on individual group members, sometimes radically changing the behavioural output observed [8, 9]. Indeed, it was previously shown that lone bees are more likely to initiate an attack against a visuo-tactile stimulus than bees within a pair, although both lone and paired bees are equally likely to react when confronting the same stimulus in the presence of the main component of the alarm pheromone [14]. Although this already suggested that individual stinging responses are affected by social context, it was not tested further mainly because of the difficulty to analyse data observations for larger group sizes (from the combinatorial explosion of possible group dynamics leading to the same final observation). By formally establishing both a model and fitting methods, the work presented here solves this issue. As a result, we were able to compare the alarm pheromone responsiveness of individuals in group sizes ranging from 1 to 15 bees. We show that, as group size increases, bees become less likely to sting in response to a given alarm pheromone concentration. Thus, it seems that the presence of nestmates inhibits stinging behaviour to some extent.

Of course, our approach has limitations that need to be kept in mind. Among them, one can cite the necessity for a definite group size and spatial homogeneity (of alarm pheromone dispersal). While both of these requirements are met in the set-up used to collect our experimental data, they are not likely to be encountered in the wild: disturbances are usually located at the nest entrance, where SAP levels may differ widely depending on entrance geometry and wind conditions. In addition, bee traffic at the entrance may fluctuate for reasons independent of the defensive situation. Our current methods also can not scale to very large population sizes such as those of a bee colony, because of the exponential number of rational functions that would need to be generated in order to describe our model. However, most attackers are repelled at the hive entrance, where only a subset of bees are present. Thus, it may be that the effective population that needs to be considered in the context of defence is not that big. The size of the predator is also an important point to consider [37], especially in the case of our small dummy we expect that the range of group sizes considered is likely appropriate. Another

important assumption for our model is that stinging likelihood as a function of SAP level is non-decreasing (but it can remain constant). This could seem at odds with the previous work on individual bees [21], in which stinging likelihood appeared to decrease at high SAP levels. An important difference, however, is that in this case the bee was confronted directly with a high concentration of SAP, whereas in the group experiments the bees experience step by step increases in SAP up to high levels. The assumption that we make, therefore, is that bees sting as soon as their threshold is reached—they do not "wait" and thus they cannot re-assess their decision after higher SAP levels are reached. This is partly supported by the fast reaction times observed when testing bees in the presence of the alarm pheromone [38]. Nonetheless, it could be that some bees change their minds, i.e. that the internal decision to sting is evaluated multiple times before any action takes place. This, however, will remain impossible to verify experimentally as long as we can only observe the behavioural output. At the colony level, a study also reported that some colonies "retreat" rather than "release" defenders in response to disturbances (mechanical shocks and/or SAP) [39], which would not be possible in our model. We note, however, that in this case the bees were not provided with a moving target, which is essential in order to trigger a flying and stinging response [40, 41].

While we acknowledge that our model is not directly transferable to a wild situation, a mechanistic understanding of the defensive behaviour of honeybees is also difficult to access via field tests because of the sheer number of factors involved in this response. Controlled laboratory assays such as ours provide a way forward, and in this setting our model successfully captures the most relevant features of this behaviour. Indeed we could use it to compare the responses of individuals when embedded within groups of different sizes. We find that recruitment is curbed as group size increases, thus demonstrating the existence of a negative social effect on stinging behaviour. Such a *per capita* decrease in aggression as a function of group size had already been described in wasps [11] and stingless bees [13]. In honeybees, it was found that bees grouped with gentle 1-day old conspecifics, who do not contribute to defence, are more aggressive than bees in older groups [42]. While this study also suggests that the social environment modulate aggressive reactions, the context considered there was defence against a non-nestmate (intruder assay). Bees rarely sting and typically do not use the alarm pheromone in this context [43], and it seems to rely on different regulation mechanisms overall [10, 44]. Our results open a number of questions for further studies. In particular, how does a bee estimate group size? Which sensory cues are being used? How are they integrated against the SAP information in the bee brain? What are the ecological consequences of such negative social feedback for colony function after an attack? We hypothesize that it may help to prevent too many bees from being recruited at the slightest disturbance, thus preserving workforce. Finally, our model also paves the way for easier comparisons between other experimental conditions (e.g. group compositions or bee species).

Our proposed model can broadly be seen as an agent-based (individual-centric) model, since the population-level behaviour emerges from local interactions of individual decision-making agents. Agent-based models come in different variants (zonal [45], force-field [46], probabilistic [22], to name a few), and are widely used in modelling collective phenomena in biology and beyond. These models link the individual's decisions to collective outcomes, and in this sense are different than models merely emulating the input-output dependencies seen at the population level. Agent-based models are typically easy to implement and simulate, hence allowing to predict the emerging population-level behaviours for a given, fixed set of parameters. However, when parameters are unknown or uncertain, inferring parameter values from population-level data measurements easily becomes challenging due to model's high dimension and stochasticity. While parameter inference for Markov (population) models is a long-studied problem, relevant in a wide range of applications including systems biology [27],

finance [20], and beyond, the existing state-of-the-art techniques are typically designed for the case when time-series experimental observations are available. We instead have dealt with measurements only at steady-state. Our proposed methodology successfully exploits the possibility to repeat the measurement at steady-state in controlled experimental conditions, coupled with a model feature—that a variety of different experimental outcomes can be observed at the steady-state. In other words, the variability in the outcomes observed at the steady-state (model containing more BSCC's) renders our parameter inference problem at hand identifiable. We expect that our approach could be easily transferable to a number of other biological systems, but it is also worth mentioning that the workflow we propose in this paper is applicable beyond the purpose of deeper understanding of biological systems: it applies to any modelling scenario where it is of interest to infer the unknown parameters in a Markov chain from repeated steady-state data measurements. Prominent examples of such scenarios arise in verification of randomised population protocols, when parameters are unknown or uncertain ('grey-box' verification scenario). For instance, in the synchronous leader election protocol [47] (https://www.prismmodelchecker.org/casestudies/synchronous_leader.php), it is of interest to verify whether a network reaches the decision after a specified number of rounds. The chain modelling such protocol after a specified number of rounds is a parametric Markov chain (branching process in this case) with two possible outcomes: either a leader has been elected, or there is the need for another round. Hence, there are two classes of BSCCs the system can finally reach. Assuming these final states can be observed repeatedly, the same methodology outlined in this paper can be applied for inferring parameters and, subsequently, verifying the property of reaching consensus within a specified number of rounds. A similar model with two BSCCs is used for other protocols like Zeroconf or randomised consensus.

## Supporting information

**S1 Text. Details of the methodology and results.** The supplementary material containing a detailed description of methods and tools employed in the methodological framework, including additional details on the obtained results and their reproducibility.
(PDF)

## Acknowledgments

MN thanks Karoline Weich, Cesar Bertinetti-Cerrato and Johanna Roller for helping to collect the experimental data, as well as Prof. Alison Mercer for hosting and encouraging the start of this study. All authors thank Dr. Elisabeth Böker, for participation in shaping the Author's summary.

## Author Contributions

**Conceptualization:** Tatjana Petrov, Morgane Nouvian.

**Data curation:** Morgane Nouvian.

**Software:** Matej Hajnal, Julia Klein.

**Supervision:** Tatjana Petrov, David Šafránek.

**Visualization:** Matej Hajnal, Julia Klein.

**Writing – original draft:** Tatjana Petrov, Matej Hajnal.

**Writing – review & editing:** David Šafránek, Morgane Nouvian.

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
