## [Decision Letter · Decision Letter 0]

1 Apr 2022

Dear Dr. Safranek,

Thank you very much for submitting your manuscript "Collective defence in honeybees:  Extracting individual behaviour from population data" for consideration at PLOS Computational Biology.

As with all papers reviewed by the journal, your manuscript was reviewed by members of the editorial board and by several independent reviewers. In light of the reviews (below this email), we would like to invite the resubmission of a significantly-revised version that takes into account the reviewers' comments. As you will see in the reports below, all the Reviewers appreciate the importance of simplified models as first steps towards more complex descriptions of biological systems, and I agree with them. Two of them, however, make important points both about the interpretation of your results and the implications that your simplified modeling approach will have in moving the field forward. Although a resubmitted version should address all the points made by all the Reviewers, I would like to encourage you to address the major comments raised by Rev. #2 and #3 about the interpretation and implications of your results very carefully.

We cannot make any decision about publication until we have seen the revised manuscript and your response to the reviewers' comments. Your revised manuscript is also likely to be sent to reviewers for further evaluation.

Sincerely,

Ricardo Martinez-Garcia

Associate Editor

PLOS Computational Biology

Natalia Komarova

Deputy Editor

PLOS Computational Biology

Reviewer's Responses to Questions

**Comments to the Authors:**

Reviewer #1: In this manuscript, the authors develop and use a new technique, using a combination of exact and statistical methods, to infer individual behaviors from only the group response. They apply this technique to study recruitment of a defense behavior in a group of honeybees. In this experiment, a fake predator was presented to a group of 10 bees, and the number of bees that stung the decoy was counted. The authors then inferred the probability that a bee initiates a stinging behavior, given a certain number of bees have already stung the decoy.

I think that the main strength of this work is in the inference technique that is developed and deployed. The general problem of inferring individual behaviors only from measurements of a collective outcome is very common in the field of collective behavior, and this methodology should prove to be very useful for many researchers. The authors have describe the methodology clearly by building up the reader's intuition by first describing a single bee, then two bees, then generalizing to an arbitrary number of bees.

I think this manuscript's contribution to our understanding of honeybees is somewhat more limited. They do extend our understanding of defense recruitment in honeybees, which may then be compared to other recruitment behaviors in honeybees, or in other species. It would have been interesting, however, to have tested other group sizes, or a larger group size. Most interesting collective behaviors occur at group sizes much larger than N = 10, so gaining some insight at those larger groups would have been useful.

On balance, I think the theoretical contributions outweigh the limited experimental contributions and would support this paper to be published in this journal.

Reviewer #2: This study proposes a modeling approach to investigate the collective attack behavior of honey bee colonies during nest defense. The argument is that it can be difficult to scale single individual behavior to collective, group-level responses. Unfortunately, I do not have the expertise to comment on the modeling approach itself, but I believe the investigators have identified an important problem, and one that could benefit from a modeling approach. I hope my comments improve the impact of this manuscript.

Given the extensive simplifying assumptions of the model (acknowledged by the authors themselves in the Discussion), I feel the value of the model and its impact are overstated. The authors state “Thanks to the new model and tools presented here, we’ll now be able to expand the study to larger group sizes, which was previously impossible.” While this may be true in the simplified arena context, does this study get us any closer to understanding what is going on inside a beehive during a predator attack?

For example, the authors highlight the importance of social context in predicting the escalation of the anti-predator response, but they do not really wrestle with what is already understood about the complexity of this response, and they do not fully justify why their approach retains value despite ignoring this complexity. For example, contrary to the model assumptions, there are a variety of studies suggesting negative, not positive social feedback in response to alarm pheromone and other defensive cues, both at the colony level and in lab-based assays (e.g., Kastberger et al. 2009, Rittschof 2017, papers with first author Hagai Shpigler). It seems like a lot of modern studies on honey bee aggression are ignored in this study. Given this (and other assumptions listed below), the model is overly simplistic. I understand that it may be a first step towards understanding this phenomenon (as mentioned in the Discussion), but the impacts of the current model seem overstated.

The authors list many critical caveats and assumptions of their model and the ways in which it fails to capture real-world biology. As a result, an informed reader is left wondering about the benefits of the model at all. To counteract this impression, the authors could do more to explain why, despite the simplicity, this modeling approach is meaningful. This should occur throughout the manuscript, not just in the Discussion. The approach would come across better if it better justified the simplifications, and perhaps gave specific examples of the ways that these could be addressed in future studies.

More detailed comments related to model simplifying assumptions that could be addressed:

How might the results of this assay in which the predator does not leave or escalate the attack track the real-world dynamics of predator response? Similarly, what are the implications of ignoring the possibility that stings may build up slowly versus quickly?

The nature of the intruder context influences whether bees show positive or negative social feedback for attack – this issue, i.e., the type of predator considered, is not clear.

L44 most defensive behaviors are low level behaviors (your data seem to support this as stinging is relatively infrequent) – please address the implications of only measuring sting response.

L84 The size of the population impacts information transfer because odor signals diffuse over physical space. How can the results with the current model be extrapolated to “any” population size, as the model ignores this component? This seems like an overstatement.

I think it would help in the Methods to contextualize the lab study with the real-world predator attack. For example, the time frame chosen was 10 min because few bees sting after this time period in the lab assay, but how does that relate to a real-world predator attack?

L158 – while the model assumptions correspond to the arena assay, they do not correspond to real-world conditions, which is the fundamental challenge to understanding collective behavior at a hive scale.

These are some extremely significant assumptions that are contradicted by real-world conditions:

-Pheromone doesn’t degrade over time

-Spatial homogeneity is assumed, which is not realistic for a bee hive.

-Bees equally influence each other, which is a simplification esp given the negative feedback that can occur

-If multiple bees react simultaneously (L168), it seems like temporal dynamics are particularly important to consider.

More should be done to justify (or just simply explain) the approach, particularly in the Methods.

L172 – individuals are known to have intrinsically different thresholds, especially across patrilines, which occur within any naturally mated colony – here you assume they are all the same. You revisit this issue in the Discussion and say that the model DOES account for different response thresholds. How or why is unclear to me – I’m not sure how to reconcile the Discussion with L172.

Other detailed comments:

L14 Unclear of purpose of this sentence.

L16 what is meant by “mechanistic understanding”?

L30 honey bees gather nectar not honey (except in extreme cases like robbing)

L35 guards can emit alarm pheromone while standing at the entrance

L103 contraction

L184 – you refer to a colony specific threshold but it is not clear to me how this is defined.

L336 – is “wrt” “with respect to”, or another acronym? Please write out.

Reviewer #3: This is an exciting topic, that many beekeepers, bee researchers, and bee enthusiasts meet: once you get stung once, there’s a good chance you’ll get stung again due to recruitment. By combining experiments and modeling, the authors show that the collective decision making is based of individuals sensing the alarm pheromone concentration. Overall, the paper is written clearly, the methods are sound, but I do have some major concerns about the interpretation of the results:

1. The analysis of the experiments is somewhat superficial, i.e., the authors only count the final number of stingers at the end of the experiment. Wouldn’t measuring the temporal value of the number of stingers provide a stronger model validation? At the moment the model validation is entirely dependent on the data presented in Fig. 2.

2. If the model cannot be better validated (point 2), the authors should at least provide some testable perditions, allowing for model validation in future experiments (e.g., predict what would happen for groups of different sizes, as the authors mention in the introduction and abstract).

3. Collection of bees: it is not clear if each experiment consisted of 10 bees from the same colony, or if they were mixed from different colony. This is an important detail, because bees from different colonies could exhibit defense response towards each other, hence altering the social dynamics of the group.

**Have the authors made all data and (if applicable) computational code underlying the findings in their manuscript fully available?**

Reviewer #1: None

Reviewer #2: Yes

Reviewer #3: Yes

PLOS authors have the option to publish the peer review history of their article (what does this mean?). If published, this will include your full peer review and any attached files.

Reviewer #1: No

Reviewer #2: No

Reviewer #3: No
---

## [Decision Letter · Decision Letter 1]

15 Jun 2022

Dear Dr. Safranek,

We are pleased to inform you that your manuscript 'Extracting individual characteristics from population data reveals a negative social effect during honeybee defence' has been provisionally accepted for publication in PLOS Computational Biology.

Best regards,

Ricardo Martinez-Garcia

Associate Editor

PLOS Computational Biology

Natalia Komarova

Deputy Editor

PLOS Computational Biology

Reviewer's Responses to Questions

**Comments to the Authors:**

Reviewer #3: The authors have done a thorough job on the revisions. I have no further comments.

**Have the authors made all data and (if applicable) computational code underlying the findings in their manuscript fully available?**

Reviewer #3: None

PLOS authors have the option to publish the peer review history of their article (what does this mean?). If published, this will include your full peer review and any attached files.

Reviewer #3: No

---

## [Editor Report · Acceptance letter]

21 Jul 2022

PCOMPBIOL-D-22-00031R1 

Extracting individual characteristics from population data reveals a negative social effect during honeybee defence

Dear Dr Šafránek,

I am pleased to inform you that your manuscript has been formally accepted for publication in PLOS Computational Biology. Your manuscript is now with our production department and you will be notified of the publication date in due course.

With kind regards,

Zsanett Szabo
